# Impact of Body Mass Index in the Cardioverter Efficacy of Amiodarone in Persistent Atrial Fibrillation

**DOI:** 10.3390/ph17060693

**Published:** 2024-05-28

**Authors:** Carmen Ligero, Pau Riera, Amine El-Amrani, Victor Bazan, José M. Guerra, Silvia Herraez, Xavier Viñolas, Josep M. Alegret

**Affiliations:** 1Servei de Cardiología, Hospital Universitari de Sant Joan, Institut d’Investigació Sanitària Pere Virgili, Grup de Recerca Cardiovascular, Universitat Rovira i Virgili, 43204 Reus, Spain; mariacarmen.ligero@salutsantjoan.cat (C.L.); amin.elamrany@gmail.com (A.E.-A.); 2Servei de Farmacia, Hospital de la Santa Creu i Sant Pau, Universitat Autònoma de Barcelona, 08025 Barcelona, Spain; priera@santpau.cat; 3Servei de Cardiología, Hospital Universitari Germans Trias i Pujol, Universitat Autònoma de Barcelona, 08916 Badalona, Spain; victorbazang@yahoo.com; 4Servei de Cardiología, Hospital de la Santa Creu i Sant Pau, Universitat Autònoma de Barcelona, 08025 Barcelona, Spain; jguerra@santpau.cat (J.M.G.); sherraez@santpau.cat (S.H.);

**Keywords:** atrial fibrillation, amiodarone, body mass index, pharmacological cardioversion, sinus rhythm

## Abstract

Background: Amiodarone is an anti-arrhythmic drug that has extensive tissue distribution and substantial storage in the fat tissue. Different studies have described some implications of body fat composition in its pharmacokinetics and pharmacodynamics. However, no clinical studies have described its implications for clinical efficacy. Methods: We studied 878 patients with persistent atrial fibrillation (AF) treated with a regimen of amiodarone and referred to electrical cardioversion (ECV), included prospectively in two Spanish registries. We analyzed the influence of body mass index (BMI), as well as overweight and obesity, in the efficacy of amiodarone for achieving pharmacologic cardioversion to sinus rhythm (SR) before ECV. Results: A total of 185 patients (21.1%) reverted to SR before ECV. Patients who reverted to SR had a lower BMI than those who did not revert (27.45 ± 4.36 kg/m^2^ vs. 29.11 ± 4.09 kg/m^2^; *p* < 0.001). We observed a progressively lower probability of reverting to SR in overweight and obese patients (normal weight 28.3%, overweight 21.3%, obesity 13.1%; *p* < 0.001). In the logistic regression, BMI (kg/m^2^) adjusted for other related variables remained as the main factor inversely related to reversion to SR (OR = 0.904 × kg/m^2^); CI 75% 0.864–0.946). Conclusions: We observed a negative relationship between an increased BMI and the efficacy of amiodarone for reversion to SR, suggesting a negative clinical impact of excess body fat in its efficacy.

## 1. Introduction

Atrial fibrillation (AF) stands as the most prevalent sustained arrhythmia, significantly contributing to elevated morbidity and mortality, primarily attributed to heart failure and stroke [1]. Previous studies from the early 2000s found no significant superiority of the rhythm control strategy over the conservative approach to heart rate control [2,3]. However, the results of more recent studies suggest a clinical benefit of the rhythm control strategy [4]. Electrical cardioversion (ECV) is commonly used in treating persistent AF as part of the rhythm control approach, along with ablation and anti-arrhythmic drugs (AAD). However, AF recurrence after ECV is notably high [5]. For this reason, the European Society of Cardiology proposes in its clinical guidelines the concomitant use of AAD with a grade IIA recommendation [6].

The most widely used AAD in Europe is amiodarone [7], which is the most effective AAD for maintaining sinus rhythm (SR). Amiodarone is mostly metabolized by the hepatic cytochrome P450 3A4 (CYP3A4) to its active metabolite desethylamiodarone and has a long half-life. Both amiodarone and its metabolite desethylamiodarone exhibit high lipophilicity [8] and extensive extravascular distribution. Their pharmacokinetic characteristics facilitate notable accumulation in adipose tissue, lungs, and liver. Concentrations of these compounds in fat compartments can range from 4 to 226 times higher than in plasma [9], surpassing even heart levels. Notably, desethylamiodarone concentrations exceed those of amiodarone in most tissues, with the exception of adipose tissue. 

Several studies have explored the implications of body fat composition on pharmacokinetics and pharmacodynamics, highlighting that fat accumulation leads to lower concentrations in plasma and the heart [10,11]. However, no clinical studies have yet detailed its clinical implications. 

These findings support the rationale for administering loading doses over several weeks in order to rapidly achieve the tissue impregnation required for therapeutic effectiveness [12]. In this context, pre-ECV administration of amiodarone could facilitate reversion to SR (pharmacological cardioversion) in approximately one-fourth of patients with persistent AF [13]. 

On the other hand, obesity plays a significant role in the development of AF through a complex interplay of structural, inflammatory, metabolic, autonomic, and epicardial adipose tissue-mediated mechanisms [14].

This study aimed to analyze how body mass index (BMI), overweight and obesity affect the clinical efficacy of amiodarone, specifically in its capacity to cardiovert patients with persistent AF, a scenario predominantly influenced by the effects of AAD.

## 2. Results

### 2.1. Patients’ Characteristics

A total of 878 patients were selected (Figure 1). Clinical and echocardiographic characteristics of the study population are shown in Table 1. Participants were predominantly male (67.5%) and the mean age was 64 ± 11 years. Most of the patients had good functional status at baseline (51% NYHA functional class I) and more than half suffered from hypertension (57.6%). With respect to AF, the prevalence of long-term AF was low (11.6%). Mean BMI was 28.78 ± 4.36 kg/m^2^, with overweight (45%) and obesity (31.2%) predominating.

### 2.2. BMI and SR Reversion

SR reversion before undergoing ECV was detected in 185 patients, with 21.1% of those included in our analysis. We observed a lower BMI among patients who reverted to SR compared to those who remained in AF (27.45 ± 4.36 kg/m^2^ vs. 29.11 ± 4.09 kg/m^2^; *p* < 0.001) (Table 1).

Among patients classified as overweight and obese, we noted a progressively lower incidence of SR reversion compared to those with normal weight (normal weight 28.3%, overweight 21.3%, obesity 13.1%; *p* < 0.001) (Figure 2).

We conducted a multivariate analysis using a logistic regression model to determine whether BMI (kg/m^2^) was independently associated with SR reversion. BMI (kg/m^2^) emerged as the primary independent variable related to SR reversion (OR = 0.90 per kg/m^2^; 95% confidence interval = 0.86–0.95; *p* < 0.001), along with other established factors associated with AF recurrence (AF duration > 1 year; presence of structural heart disease) (Table 2).

When we performed the same analysis excluding patients with structural heart disease, the results remained consistent (27.45 ± 4.36 kg/m^2^ in patients with SR reversion vs. 29.53 ± 4.38 kg/m^2^ in patients without SR reversion; *p* < 0.001; Appendix A). Similarly, when considering BMI as a categorical variable, the results remained consistent (normal weight 33.8%, overweight 26.6%, obesity 13.6%; *p* < 0.001) (Figure 2), and BMI (kg/m^2^) remained an independent variable associated with SR reversion in logistic regression (OR = 0.91; 95% confidence interval = 0.85–0.97; *p* < 0.001) (Appendix A).

## 3. Discussion

We observed an inverse relationship between BMI and the efficacy of amiodarone in reverting AF to SR in patients with persistent AF who were treated with amiodarone prior to ECV. Elevated BMI reduced the probability of reverting to SR, with a progressively lower probability observed in overweight and obese patients compared to those with normal weight. As far as we know to date, this study is the first to demonstrate a significant association between increased BMI and the efficacy of amiodarone for reverting to SR.

The diminished response observed in overweight and obese patients in the present study may be partially attributed to the pharmacokinetic characteristics of amiodarone and its main metabolite, desethylamiodarone. Lipophilic molecules such as these are predisposed to exit the bloodstream and accumulate in regions with elevated lipid density. Consistently, several studies have demonstrated the extensive accumulation of amiodarone and its metabolite in adipose tissue [10,11]. This extravascular distribution leads to an increased volume of distribution, prolonged elimination half-life, and reduced plasma concentration. Consequently, it is reasonable to hypothesize that overweight and obese patients may necessitate higher doses of amiodarone to achieve equivalent cardiac outcomes, given their greater proportion of adipose tissue.

It has been previously reported that lipophilic drugs, like amiodarone, which distribute partly into adipose tissue, should be dosed based on ideal body weight along with a percentage of the patient’s excess body weight [15]. Amiodarone is metabolized in the liver to desethylamiodarone, an active metabolite that shares properties with the parent drug but has a significantly longer elimination half-life [8]. In animal experimentation models with obese male rats, decreases in the expression of several proteins involved in drug metabolism by the liver and kidney have been reported. This suggests the inefficient metabolism of certain drugs in obese patients [16]. However, a study conducted on a Japanese population revealed alterations in the pharmacokinetics of orally administered amiodarone, with a decrease in drug clearance among patients categorized as overweight or obese based on their BMI [17]. Nonetheless, there might be undisclosed racial or ethnic variances affecting the pharmacokinetics of amiodarone. In addition, results of other studies not only indicate that obesity plays a role in mediating the response to AADs for AF but also underscore a reduced therapeutic efficacy of sodium channel blockers compared to potassium channel blocker AAD [18]. Subsequent studies should analyze whether an increase in the dose of amiodarone in obese patients is related to a clinical benefit.

BMI may influence amiodarone pharmacokinetics and modify the effect of this drug. In other drugs such as digoxin, which is used to modulate ventricular response in AF, monitoring plasma levels can be useful in certain clinical scenarios. In relation to amiodarone, several studies have examined its pharmacokinetics, but the available evidence remains limited. Indeed, amiodarone pharmacokinetics has been described using one, two, three, or even four compartment models [19]. Furthermore, therapeutic drug monitoring should include the amiodarone metabolite N-desethylamiodarone, which is thought to be pharmacologically active. Additionally, the long half-life of amiodarone presents a challenge, requiring considerable time to reach steady-state concentrations. Further research is necessary to evaluate the utility of monitoring both amiodarone and its main active metabolite in clinical practice.

We recently found that in patients with persistent AF, a higher BMI is associated with an increased likelihood of AF recurrence following ECV [20]. In this clinical scenario, beyond considering the pharmacokinetic and pharmacodynamic properties of amiodarone, other obesity-related factors implicated in AF recurrences may also play a significant role, such as obesity-associated sleep disorders. In obstructive sleep apnea syndrome, the emergence of nocturnal hypoxia and acidosis can act as triggers favoring AF recurrence [21,22]. Conversely, pharmacological cardioversion in persistent AF mainly depends on the efficacy of amiodarone and its capacity for cardioversion, along with its distribution in adipose tissue. Likewise, the duration of the arrhythmia is also an important factor. Persistent AF is defined as AF lasting longer than seven days, necessitating cardioversion to terminate episodes due to the low probability of spontaneous reversion to SR [23]. Therefore, it can be inferred that the restoration of SR primarily results from the pharmacological effect of amiodarone.

In addition, obesity is associated with the enlargement of cardiac chambers, including atrial remodeling [24,25] a phenomenon linked to both the onset and persistence of AF [26,27], and potentially correlated with a reduced likelihood of pharmacological reversion to SR. However, it is noteworthy that even after excluding patients with structural heart disease and adjusting for other relevant variables, such as atrial size, overweight and obesity persisted as factors associated with a diminished probability of SR reversion.

Moreover, there is a well-established correlation between BMI and epicardial fat [28]. Epicardial fat has been suggested as a potential contributor to AF pathophysiology. This is because epicardial adipose tissue exhibits endocrine activity that may promote fibrosis of the atrial myocardium. The suggested mechanism involves the paracrine effect of adipocytokines released by epicardial adipose tissue (EAT), amplified by myocardial tissue infiltration by fat and inflammation, ultimately leading to fibrotic restructuring of adipose tissue in the atrial epicardium [29]. The emerging understanding of EAT’s multifaceted contributions to AF pathophysiology underscores its potential as a therapeutic target for innovative AF management strategies [30]. Future research efforts focusing on EAT-directed interventions hold promise for advancing personalized and effective approaches to AF.

Obesity is intricately linked to the development and progression of AF through various mechanisms, including metabolic dysfunction, inflammation, and neurohormonal changes. Insulin resistance, dyslipidemia, and hyperglycemia associated with obesity promote atrial fibrosis, oxidative stress, and mitochondrial dysfunction, creating an arrhythmogenic substrate for AF. Inflammation, both at the adipose tissue and systemic levels, contributes to atrial electrical and structural remodeling, facilitating AF initiation and perpetuation. Additionally, sympathetic nervous system overactivity and activation of the renin-angiotensin-aldosterone system promote atrial electrical instability and fibrosis, predisposing obese individuals to AF. These interrelated mechanisms illustrate the complexity of obesity’s role in AF development and underscore the importance of therapeutic approaches aimed at mitigating these pathological processes in the prevention and management of AF in obese patients [31,32].

We must acknowledge several limitations of our study. Firstly, BMI serves as a simplistic measure for evaluating overweight or obesity, and body fat may be estimated by formulae derived. Other anthropometric methods, such as waist–hip ratio or skinfold estimation, or specific techniques such as plethysmography or dual-energy X-ray absorptiometry would be more precise [33,34], but less affordable. Additionally, we cannot dismiss the possibility that undiagnosed and so untreated obstructive sleep apnea [35] may also affect the response to amiodarone in overweight and obese AF patients, potentially reducing the likelihood of SR reversion [36]. 

## 4. Materials and Methods

### 4.1. Study Population

Data were obtained from two Spanish registries focused on monitoring the clinical practice of ECV in patients with persistent AF [37,38]. These registries enrolled prospectively and consecutively patients from 99 Spanish hospitals with persistent AF for whom an ECV was indicated. Inclusion criteria were uniform across the registries, requiring patients to be over 18 years old with persistent AF, defined as an arrhythmia lasting seven or more days without precipitating conditions such as hyperthyroidism, fever, recent thoracic surgery, or pericarditis. Recorded data encompassed clinical characteristics, treatment specifics, echocardiography results, and ECV procedure variables. 

From a total of 2430 ECV candidates across both registries, we selected those patients treated with an amiodarone regimen initiated one month prior to ECV as AAD (beta-blockers or calcium antagonists were allowed). The recommended regimen involved 600 mg daily for the first week, 400 mg daily for the second week, and 200 mg daily until ECV. Pharmacological cardioversion induced by amiodarone was considered when sinus rhythm (SR) was detected on an electrocardiogram (ECG) obtained before performing ECV. Although monitoring the QT interval is mandatory during treatment with amiodarone, this parameter was not collected in the present study database. Similarly, the analytical parameters for evaluating thyroid function were also not collected. In both registries, those patients in whom AAD was withdrawn were documented, along with the reasons for withdrawal (such as toxicity, other adverse effects, ineffectiveness, etc.). Amiodarone was not discontinued before ECV in any case. Structural heart disease was defined by the presence of anomalies such as moderate or severe valvular heart disease, any grade of mitral stenosis, previous myocardial infarction, systolic dysfunction (ejection fraction < 50%), or any cardiomyopathy [38]. Those patients who did not revert to SR underwent ECV. The flow chart is presented in Figure 1.

Weight categories were established by BMI according to World Health Organization (WHO) criteria [39]: normal weight was defined as a BMI between 18.5 and 24.9 kg/m^2^, overweight as a BMI between 25 and 29.9 kg/m^2^, and obesity as a BMI equal to or greater than 30 kg/m^2^. Subgroups of obesity (classes 1, 2, and 3) were not considered due to the significant reduction in the number of patients per group. 

### 4.2. Statistical Analysis

Continuous quantitative variables with a normal distribution are presented as mean ± standard deviation (SD), while categorical variables are expressed as frequencies and percentages.

The comparison between patients who reverted to SR and those who did not revert was conducted using a Chi-square test.

Correlations between BMI and SR reversion in the subgroups were assessed using the Kendall rank correlation coefficient. Univariable and multivariable regression analyses were performed to evaluate predictors of SR reversion (logistic regression), with a specific focus on BMI considered both as a quantitative variable and as a variable categorized into normal weight, overweight, and obesity categories.

Additionally, a stratified analysis was carried out based on the WHO definition [39], categorizing patients into different BMI groups to compare overweight and obese individuals (BMI 25–29.9 and BMI ≥ 30 kg/m^2^, respectively) with those classified as having a normal BMI (18.5–24.9 kg/m^2^).

Statistical significance was defined as a *p*-value < 0.05.

All statistical analyses were performed using IBM SPSS Statistics for Windows, Version 29.0.

Studies were conducted in accordance with the principles outlined in the Declaration of Helsinki and were approved by the Institutional Review Board and Ethics Committee of the Hospital Universitari de Sant Joan (main board; code 11-12-22/12obs3) and subsequently by all participating centers. All participants provided informed written consent before taking part in the studies.

## 5. Conclusions

In our study, we found that higher BMI levels within the obesity and overweight range were linked to reduced effectiveness of amiodarone in reverting to SR among patients with persistent AF. This suggests a detrimental clinical effect of excess body fat on amiodarone’s efficacy.

Overall, we think that our study provides valuable insights into the complex relationship between obesity, pharmacological treatment efficacy, and AF management, underscoring the importance of individualized approaches in clinical practice.

## Figures and Tables

**Figure 1 pharmaceuticals-17-00693-f001:**
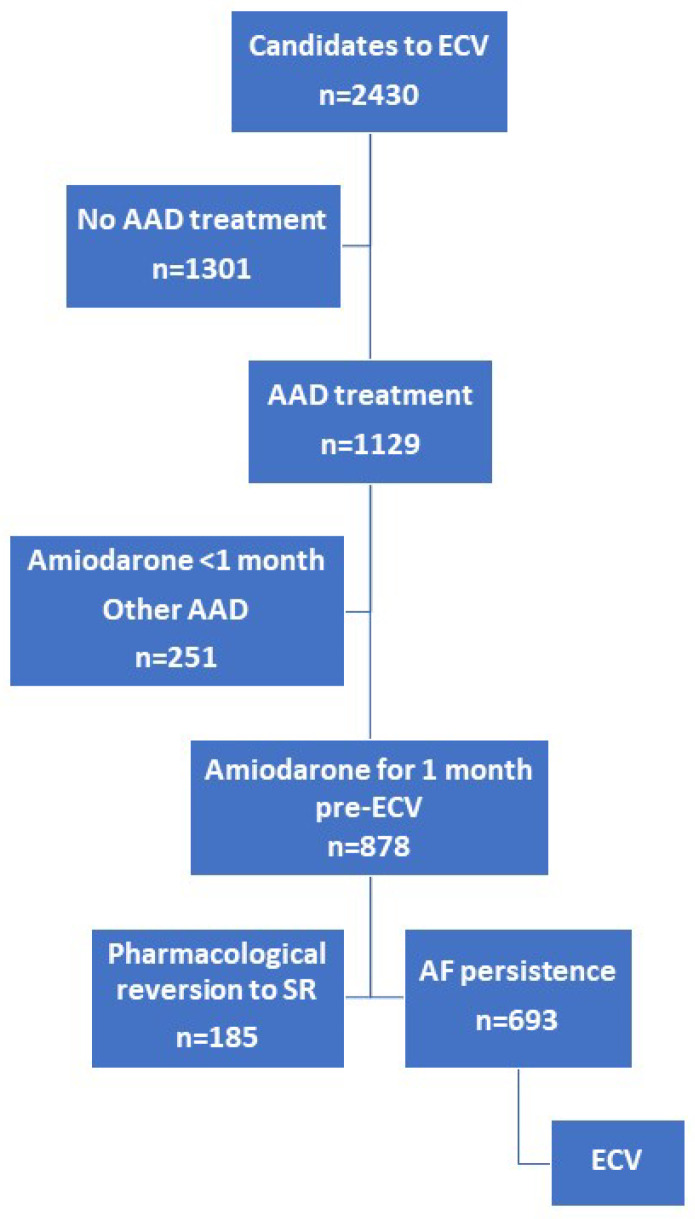
Flow chart representing patient inclusion. AF, atrial fibrillation; ECV, electrical cardioversion; SR, sinus rhythm; AAD, anti-arrhythmic drugs.

**Figure 2 pharmaceuticals-17-00693-f002:**
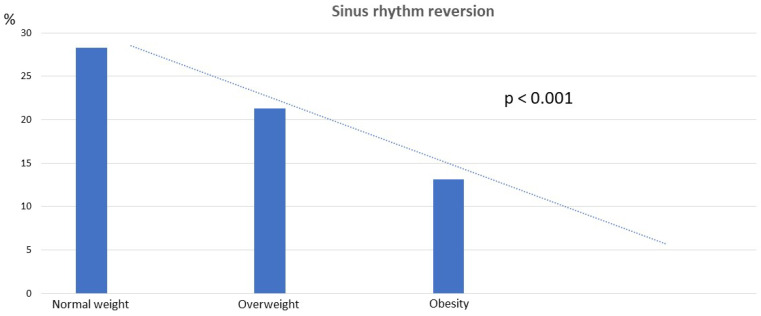
Sinus rhythm reversion by BMI categories.

**Table 1 pharmaceuticals-17-00693-t001:** Baseline characteristics of the 878 patients included.

	n = 878 (%)	No PCV (n = 693)	PCV (n = 185)	*p*
Age (years)	64 ± 11	63.99 ± 10.92	64.99 ± 11.33	0.150
Male gender	593 (67.5)	482 (69.5)	111 (60.0)	0.014
Structural heart disease	411(46.8)	337 (48.6)	74 (40.0)	0.037
Diabetes mellitus	130 (14.8)	108 (15.6)	22 (11.9)	0.209
Hypertension	506 (57.6)	403 (58.2)	103 (55.7)	0.545
COPD	80 (9.1)	65 (9.4)	15 (8.1)	0.590
LVH	309 (36.0)	253 (36.5)	56 (30.3)	0.101
LVEF < 40%	89 (10.9)	70 (10.0)	19 (10.7)	0.910
NYHA ≥ 2	429 (48.9)	338 (48.8)	91 (49.3)	0.908
AF duration > 1 year	102 (11.6)	90 (12.9)	12 (6.7)	0.014
LA size > 50 mm	162 (19.7)	136 (19.4)	26 (14.6)	0.066
LA size	44.75 ± 6.40	45.18 ± 6.34	43.15 ± 6.38	<0.001
LVEF	57.83 ± 12.16	57.55 ± 12.19	58.87 ± 12.03	0.205
ACE/ARA II	513 (58.4)	406 (58.6)	107 (57.8)	0.854
Beta blockers	564 (64.2)	445 (64.2)	119 (64.3)	0.98
Calcium antagonist	186 (21.2)	35 (18.9)	151 (21.8)	0.40
BMI	28.78 ± 4.36	29.11 ± 4.09	27.45 ± 4.36	<0.001

AF, atrial fibrillation; BMI, body mass index; COPD, chronic obstructive pulmonary disease; LA, left atrium; LVEF, left ventricular ejection fraction; LVH, left ventricle hypertrophy; NYHA, New York Heart Association; PCV, pharmacological cardioversion; No PCV, reversion to sinus rhythm not achieved.

**Table 2 pharmaceuticals-17-00693-t002:** Parameters associated with reversion to SR before ECV in patients treated with amiodarone. Multivariate analysis.

	OR (95% CI)	*p*
BMI	0.904 (0.864–0.946)	<0.001
Previous heart disease	0.590 (0.412–0.846)	0.004
AF duration > 1 year	0.417 (0.204–0.855)	0.017
Gender		0.083
LA size (mm)		0.432

AF, atrial fibrillation; BMI, body mass index; ECV, electrical cardioversion; SR, sinus rhythm; LA, left atrial.

## Data Availability

The data presented in this study are available on request from the corresponding author. The data are not publicly available due to privacy restrictions.

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
