# Peer review of "Impact of Body Mass Index in the Cardioverter Efficacy of Amiodarone in Persistent Atrial Fibrillation"

_pharmaceuticals, 2024, doi:10.3390/ph17060693_

Round 1
Reviewer 1 Report
Comments and Suggestions for Authors
Thank you for the opportunity to review the paper: Impact of body mass index in the cardioverter efficacy of amiodarone in persistent atrial fibrillation
In my opinion, the manuscript does not qualify for publication and for further improvement due to incorrect assumptions. BMI cannot be a parameter that completely evaluates the behavior of amiodarone in tissues.
The authors themselves point this out in Limitations. In my opinion, it is not possible to draw conclusions based on BMI alone
Conclusion:
In our study, an increased BMI in the range of obesity and overweight was associated with a lower efficacy of amiodarone for reversion to SR in patients with persistent AF, suggesting a negative clinical impact of excess body fat in its efficacy. Based on our results, we postulate that high BMI should be taken into account in the dose of amiodarone prescribed.
This is an extremely high simplification and cannot be used in scientific papers
Author Response
Thank you for your honest assessment of our manuscript. We have eliminated the sentence “Based on our results, we postulate that high BMI should be taken into account in the dose of amiodarone prescribed” in the Conclusions. On the other hand, in the new version we deepened in the Discussion on the relationship between BMI and the effectiveness of amiodarone.
Reviewer 2 Report
Comments and Suggestions for Authors
A very important issue from a practical point of view. The authors conducted an interesting study, with an appropriate introduction (notes below). Results presented properly, well discussed.
A few comments that occurred to me while reading the manuscript
- patients were qualified for amiodarone treatment. Why was no decision made about electrical cardioversion? I understand that the patient was stable, but was the patient informed about the potential consequences of waiting a month for CV, or did he decide not to perform ECV?
- body weight dependence is a typical example of pharmacokinetic dependence modifying the effect. Mamay currently has publications summarizing therapy with monitored dogoxin or carbamazepine over a 20-year follow-up period. Amiodarone has never been treated as a typical drug for TDM, which does not mean that it is not possible. Please mention the TDM treatment and the possible monitoring of amiodarone.
- if we are talking about TDM, then another question - QT is assessed during amiodarone treatment. How has it changed over the course of treatment in populations? was the change in elongation different in the study groups?
- how did TSH and nt-pro-BNP behave?
- what treatment apart from amiodarone was used? was it identical in the assessed populations?
- what was the physical activity of these people? did they practice sports that increased the risk of developing FA?
-if and what were the risk factors for FA occurrence in the assessed populations?
conclusions should be more practical and not a simple summary of results
Author Response
A very important issue from a practical point of view. The authors conducted an interesting study, with an appropriate introduction (notes below). Results presented properly, well discussed. A few comments that occurred to me while reading the manuscript:
- patients were qualified for amiodarone treatment. Why was no decision made about electrical cardioversion? I understand that the patient was stable, but was the patient informed about the potential consequences of waiting a month for CV, or did he decide not to perform ECV?
Patients included in these registries were patients with persistent AF referred for elective ECV. Consequently, patients treated in the emergency department who required electrical cardioversion were not included. Among all included patients, we selected those patients whose amiodarone treatment had been initiated one month before ECV. This form of slow oral impregnation has the objective that the patient is impregnated with amiodarone at the time of ECV and is the most used in Spain. In this regimen of oral amiodarone impregnation, a progressive weekly increase in dose is usually used. On the other hand, this month of treatment gives the possibility that pharmacological cardioversion induced by amiodarone may occur. In this analysis we excluded those patients treated with shorter amiodarone regimens to have a homogeneous sample regarding the possibility of this pharmacological cardioversion occurring. As we commented in the Methods section, the patients included were treated within routine clinical practice, so the indication and duration of antiarrhythmic treatment were the decision of the cardiologist. In any case, this decision was not determined by the study protocol. This month of waiting is not related to a worse prognosis. In fact, 21% of patients did not require ECV because they had a PCV. Those who did require ECV were impregnated with amiodarone, which reduces the AF recurrence at the most critical moment, the first month after ECV.
Following the reviewer's comment and to avoid misunderstandings, we have introduced a flow chart and the following phrase in Methods section: “Those patients who did not revert to SR underwent ECV” .
- body weight dependence is a typical example of pharmacokinetic dependence modifying the effect. Mamay currently has publications summarizing therapy with monitored dogoxin or carbamazepine over a 20-year follow-up period. Amiodarone has never been treated as a typical drug for TDM, which does not mean that it is not possible. Please mention the TDM treatment and the possible monitoring of amiodarone.
We have included the next comment in the Discussion: “BMI may influence amiodarone pharmacokinetics and modify the effect of this drug. In other drugs such as digoxin, which is used to modulate ventricular response in AF, monitoring plasma levels can be useful in certain clinical scenarios. In relation to amiodarone, several studies have examined its pharmacokinetics, but the available evidence remains limited. Indeed, amiodarone pharmacokinetics has been described using one, two, three, or even four compartment models. (22). Furthermore, therapeutic drug monitoring should include the amiodarone metabolite N-desethylamiodarone, which is thought to be pharmacologically active. Additionally, the long half-life of amiodarone presents a challenge, requiring considerable time to reach steady-state concentrations. Further research is necessary to evaluate the utility of monitoring both amiodarone and its main active metabolite in clinical practice.”
- if we are talking about TDM, then another question - QT is assessed during amiodarone treatment. How has it changed over the course of treatment in populations? was the change in elongation different in the study groups?
Unfortunately, this variable (QT interval duration) was not recorded jn these registries.
- how did TSH and nt-pro-BNP behave?
As in the case of QT interval duration, these variables were not recorded. However, amiodarone was not withdrawn to any patient due to hyper- or hypothyroidism during the month of treatment prior to ECV.
- what treatment apart from amiodarone was used? was it identical in the assessed populations?
We have included in Table 1 treatment with ACE/ARA II, beta blockers and calcium antagonist. The distribution was similar in both groups (PCV vs No PCV).
- what was the physical activity of these people? did they practice sports that increased the risk of developing FA?
Unfortunately, the grade of physical activity was not recorded. We recorded the main variables that could be related to the objectives of the registries.
-if and what were the risk factors for FA occurrence in the assessed populations?
conclusions should be more practical and not a simple summary of results
The main risk factors for AF are described in Table 1. We have modified the Conclusions section following the reviewer’s recommendation.
We thank this reviewer, whose comments have allowed us to improve our manuscript.
Reviewer 3 Report
Comments and Suggestions for Authors
Currently, expanding epicardial adipose tissue in obese patients has been suggested as a pivotal factor involved in atrial fibrillation (AF) development through paracrine signalling and direct infiltration. Weight loss has been shown to decrease the AF risk and its recurrences after ablation. Previously studies have shown that a decreased hepatic metabolism and an increased level of amiodarone‐binding proteins may contribute to altered amiodarone pharmacokinetics in obese patients. In addition, amiodarone clearance has been demonstrated to be affected by BMI, which is reduced in obese patients.
Based on these data, to improve the quality of the manuscript I recommend the following to the authors:
- Please revise the text in line 52. It is hard to read.
- The authors should add references regarding data presented in lines 58 and 185.
- Kindly expand the Introduction section with data regarding the role of obesity in atrial fibrillation development.
- Kindly add more data regarding the mechanisms of obesity in AF - metabolic dysfunction, inflammation, and neurohormonal changes in the Discussion Section (lines 202-208).
- Moreover, the authors should consider discussing the previous results, which have been published by Ornelas‐Loredo et al. In their study, the therapeutic response to class III AADs (including amiodarone) was similar between obese and healthy‐weight patients. (Fukuchi H et al., 2009; doi: 10.1111/j.1365-2710.2008.00987.x
Abdussalam A et al., 2017; doi: 10.1016/j.xphs.2017.02.002
Ornelas‐Loredo A et al., 2020; doi: 10.1001/jamacardio.2019.451)
- The epicardial adipose tissue plays a central role, providing an attractive target for novel therapies in AF. The authors should discuss this aspect in the Discussion Section.
- Please revise the reference list according to the Pharmaceuticals Journal recommendation.
Author Response
- Please revise the text in line 52. It is hard to read.
Thank you for pointing this out. We agree with this comment. Therefore, we have made some modifications to the text to improve its quality and understanding.
- The authors should add references regarding data presented in lines 58 and 185.
Thank you for pointing this out. In the new manuscript we have properly placed the references (10,11) and (19,20) in the cited text.
- Kindly expand the Introduction section with data regarding the role of obesity in atrial fibrillation development.
Thank you for pointing this out. We agree with this comment. We have, accordingly, modified the introduction text to emphasize this point: “On the other hand, obesity plays a significant role in the development of AF through a complex interplay of structural, inflammatory, metabolic, autonomic, and epicardial adipose tissue-mediated mechanisms (14).”
- Kindly add more data regarding the mechanisms of obesity in AF – metabolic dysfunction, inflammation, and neurohormonal changes in the Discussion Section (lines 202-208).
Thank you for pointing this out. We have modified the Discussion to emphasize this point. We have introduced the next text: “Obesity is intricately linked to the development and progression of AF through various mechanisms, including metabolic dysfunction, in-flammation, and neurohormonal changes. Insulin resistance, dyslipidemia, and hyperglycemia associated with obesity promote atrial fibrosis, oxidative stress, and mitochondrial dysfunction, creating an arrhythmogenic substrate for AF. Inflammation, both at the adipose tissue and systemic levels, contributes to atrial electrical and structural remodeling, facilitating AF initiation and perpetuation. Additionally, sympathetic nervous system overactivity and activation of the renin-angiotensin-aldosterone system promote atrial electrical instability and fibrosis, predisposing obese individuals to AF. These interrelated mechanisms illustrate the complexity of obesity's role in AF develop-ment and underscore the importance of therapeutic approaches aimed at mitigating these pathological processes in the prevention and management of AF in obese patients (36,37).”
- Moreover, the authors should consider discussing the previous results, which have been published by Ornelas‐Loredo et al. In their study, the therapeutic response to class III AADs (including amiodarone) was similar between obese and healthy‐weight patients. (Fukuchi H et al., 2009; doi: 10.1111/j.1365-2710.2008.00987.x Abdussalam A et al., 2017; doi: 10.1016/j.xphs.2017.02.002 Ornelas‐Loredo A et al., 2020; doi: 10.1001/jamacardio.2019.451)
Thank you for your valuable contributions. We have modified the Discussion section in this sense and added the new references: “Amiodarone is metabolized in the liver to desethylamiodarone, an active metabolite that shares properties with the parent drug but has a significantly longer elimination half-life (8). In animal experimentation models with obese male rats, decreases in the expression of several proteins involved in drug metabolism by the liver and kidney have been reported. This suggests inefficient metabolism of certain drugs in obese patients. (19). However, a study conducted on a Japanese population revealed alterations in the pharmacokinetics of orally administered amiodarone, with a decrease in drug clearance among patients categorized as overweight or obese based on their BMI (20). Nonetheless, there might be undisclosed racial or ethnic variances affecting the pharmacokinetics of amiodarone. In addition, results of other studies not only indicate that obesity plays a role in mediating the response to AADs for AF but also underscore a reduced therapeutic efficacy of sodium channel blockers compared to potassium channel blocker AAD (21). Sub-sequent studies should analyze whether an increase in the dose of amiodarone in obese patients is related to a clinical benefit.”
- The epicardial adipose tissue plays a central role, providing an attractive target for novel therapies in AF. The authors should discuss this aspect in the Discussion Section.
Thank you for pointing this out. We have modified the Discussion section to emphasize this point: “The emerging understanding of EAT’s multifaceted contributions to AF pathophysiology underscores its potential as a therapeutic target for innovative AF management strategies (35). Future research efforts focusing on EAT-directed interventions hold promise for advancing personalized and effective approaches to AF.”
Reviewer 4 Report
Comments and Suggestions for Authors
The manuscript tried to demonstrate how BMI affects cardioverter efficacy of amiodarone in persistent atrial fibrillation. Although this topic is interesting, the author prepared the manuscript casually, with considerable mistakes in data collection, presentation, and interpretation.
1. Please maintain consistency in font size throughout the manuscript
2. Why do the selected patients need to be over 18 years old? The rationale?
3. Line 85, why previous myocardial infarction, systolic dysfunction (EF <50%), or any cardiomyopathy are considered as structural heart disease? What are their relationships with the research purpose of this article?
4. Please define WHO, BMI, et al when they first appear.
5. Why a total of 878 patients were selected? why the other patients (2430 candidates) were excluded from the study? Please draw a flowchart on how to exclude other patients?
6. Table 1, does PCV mean that only amiodarone was used, or combined with other AAD? Please specify.
7. Line 130-133, table 1 only shows that the 185 patients who were treated with PCV are with a lower BMI that of the 693 patients who were not treated with PCV, how the authors observed a lower BMI among those patients who reverted to SR compared to those who remained in AF? Please check carefully. If this information is wrong, the subsequent analysis were all wrong.
8. Fig.1 is missing
Comments on the Quality of English LanguageEnglish use is fine.
Author Response
The manuscript tried to demonstrate how BMI affects cardioverter efficacy of amiodarone in persistent atrial fibrillation. Although this topic is interesting, the author prepared the manuscript casually, with considerable mistakes in data collection, presentation, and interpretation.
- Please maintain consistency in font size throughout the manuscript
Thank you for your observation. In the new version we use the same font size throughout the manuscript.
- Why do the selected patients need to be over 18 years old? The rationale?
We clarify this point. Being over 18 years old is an usual selection criteria due to ethical questions and the different clinical characteristics of children and adolescents. Morover, it is exceptional that there is a patient under 18 years of age who requires elective ECV due to persistent AF. Even more so in the absence of recent cardiac surgery, which is described as an exclusion criteria.
- Line 85, why previous myocardial infarction, systolic dysfunction (EF <50%), or any cardiomyopathy are considered as structural heart disease? What are their relationships with the research purpose of this article?
The presence of any cardiomyopathy, heart valve disease, myocardial infarction or LV systolic dysfunction from any cause are considered usually structural heart disease. The presence of structural heart disease favors the appearance of AF and AF recurrences. Furthermore, it could be related with lower probabilities of pharmacological cardioversion. By these reasons, it was included in this analysis.
- Please define WHO, BMI, et al when they first appear.
Thank you for you for your observation. In the previous version BMI was defined only in the abstract. In the new version BMI is described in its first appear (Introduction; page 2, line 71) in the manuscript.
WHO is defined in Methods; page 3, line 100.
- Why a total of 878 patients were selected? why the other patients (2430 candidates) were excluded from the study? Please draw a flowchart on how to exclude other patients?
Thanks to the reviewer for his/her comment. We note that the description of the selection of participants is not entirely clear. As we describe in the text, of all the patients included in the registries (2430 patients), we selected those who had been treated with an amiodarone impregnation regimen started one month before the ECV. We also applied the usual exclusion criteria for secondary causes of AF (see Methods section), finally resulting in 878 that met these criteria. In this new version we have drawn a flowchart and we have introduced a sentence in the Methods section to define better the participants in the study (page 3, line 89).
- Table 1, does PCV mean that only amiodarone was used, or combined with other AAD? Please specify.
Effectively, we selected those patients treated with amiodarone as the only AAD (except beta blockers or calcium antagonists). In the new version this is specified in page 3, line 89. We also describe the meaning of “PCV” and “No PCV” at the bottom of Table 1.
- Line 130-133, table 1 only shows that the 185 patients who were treated with PCV are with a lower BMI that of the 693 patients who were not treated with PCV, how the authors observed a lower BMI among those patients who reverted to SR compared to those who remained in AF? Please check carefully. If this information is wrong, the subsequent analysis were all wrong.
Thanks for the comment. Table 1 shows that the 185 patients who reverted to sinus rhythm (PCV) had a lower BMI than those who did not revert (no PCV). As we said in the point before, to avoid misunderstandings, in the new version we describe “PCV” and “non-PCV” at the bottom of Table 1: PCV, pharmacological cardioversion; No PCV, reversion to sinus rhythm not achieved with amiodarone.
- Fig.1 is missing
Thank you for identifying this mistake. Figure 1 now is Figure 2 and it is presented in page 4.
We thank this reviewer, whose comments have allowed us to improve our manuscript.
Reviewer 5 Report
Comments and Suggestions for Authors
The authors investigated the association between body mass index (BMI) and the efficacy of amiodarone for achieving pharmacologic cardioversion to sinus rhythm in 878 patients with persistent AF. However, I have some comments.
1) In Table 1, please describe the duration (mean or median value) of AF in addition to the prevalence of AF duration >1 year.
2) Regardiing the treatment of AF, please descriibe the percentages of patients taking beta-blockers or digosin.
3) In Table 1, please describe the history of current smoking and alcohol drinking.
4) I would like to know blood amiodarone levels in patients with and without reversion on ECV. Was BMI a better predictor associated with reversion to SR than blood amiodarone levels?
5) I recommend the authors to describe body weight (BW) and the waist-hip ratio in Table 1. Moreover, was the BMI a better predictor associated with the reversion to SR than BW or the waist-hip ratio?
Author Response
The authors investigated the association between body mass index (BMI) and the efficacy of amiodarone for achieving pharmacologic cardioversion to sinus rhythm in 878 patients with persistent AF. However, I have some comments.
- In Table 1, please describe the duration (mean or median value) of AF in addition to the prevalence of AF duration >1 year.
Thank you for your comment. AF duration was recorded as a qualitative variable. We must take into account that 18% of the patients had an indefinite duration of AF. We selected the duration of PA that showed more significant differences between the two groups.
2) Regarding the treatment of AF, please describe the percentages of patients taking beta-blockers or digosin.
We included those patients treated with amiodarone and beta blockers or calcium antagonist. The number of patients treated with digoxin was very low. To avoid bias due to increased toxicity, we did not include this scare number of patients treated with amiodarone and digoxin. It is now specified in the methods section (page 3, line 89) and the patients treated with beta blockers and calcium antagonists are presented in Table 1.
3) In Table 1, please describe the history of current smoking and alcohol drinking.
Unfortunately, we did not collect these variables.
4) I would like to know blood amiodarone levels in patients with and without reversion on ECV. Was BMI a better predictor associated with reversion to SR than blood amiodarone levels?
Amiodarone levels are not usually used in clinical practice. These registries were performed under conditions of routine clinical practice. We are sorry but have not this data. However, we discuss now its possible usefulness in the Discussion section.
5) I recommend the authors to describe body weight (BW) and the waist-hip ratio in Table 1. Moreover, was the BMI a better predictor associated with the reversion to SR than BW or the waist-hip ratio?
The registries that are part of this study did not include the waist-hip as a variable. We have expanded a comment in the discussion regarding the limitation of having used BMI as the only variable related to obesity and body fat: “Firstly, BMI serves as a simplistic measure for evaluating overweight or obesity, and body fat may be estimated by formulae derived. Other antropometric methods, such as waist-hip ratio or skinfold estimation, or specific techniques such as plethysmography or dual energy X-ray absorptiometry would be more precise (38,39), but less affordable.”
We thank this reviewer, whose comments have allowed us to improve our manuscript.
Round 2
Reviewer 1 Report
Comments and Suggestions for Authors Thank you very much for the changes you made.The manuscript in its current form meets the criteria for publication
Author Response
Thank you for your comments.
Reviewer 2 Report
Comments and Suggestions for Authors
The authors make corrections. Some of the comments could not be applied due to the fact that the authors did not collect such data. I understand this fact, but the assessment of QT changes during aniodarone treatment is mandatory. The doctor must carry out such an assessment during visits - it is necessary to assess safety. Similarly, TSH controls at baseline and during treatment... For these reasons, these data are most likely available in the patient's documentation, but were not collected in the database of the authors of the manuscript. In the introduction, the authors must note that these data have not been evaluated, which does not change the fact that such evaluations must be carried out! Such information must be included both in the introduction (methods of monitoring amiodarone therapy) and in the methodology.
Author Response
In the new version we introduce the reviewer's suggestion (highlighted in yellow). Likewise, we introduce the comment that refers to the fact that, in both registries, those patients in whom AAD was withdrawn were documented, along with the reasons for withdrawal (such as toxicity, other adverse effects, ineffectiveness, etc.). Amiodarone was not discontinued before ECV in any case.
Thank you for your review
Reviewer 3 Report
Comments and Suggestions for Authors
The manuscript has been improved by responding to comments and integrating missing references. I have only a few minor suggestions for the authors:
- Kindly improve the typographical quality of Figure 1 (Flow chart representing patients inclusion criteria)
- Please revise the references list according to the Pharmaceuticals journal recommendation
https://www.mdpi.com/journal/pharmaceuticals/instructions
Author Response
We have followed your recommendations.
Thank you for your review
Reviewer 4 Report
Comments and Suggestions for Authors
The authors have addressed my concerns with a significantly improved manuscript. I have no further comments.
Author Response
Thank you for your review
Reviewer 5 Report
Comments and Suggestions for Authors
I have no further comments.
Author Response
Thank you for your review